# $\mu$KE: Matryoshka Unstructured Knowledge Editing of Large Language Models

**Zian Su[1]\*, Ziyang Huang[2]\*, Kaiyuan Zhang[1], Xiangyu Zhang[1]**

[1]Purdue University, [2]Johns Hopkins University

{su284,zhan4057}@purdue.edu, zhuang86@jhu.edu, xyzhang@cs.purdue.edu

## Abstract

Large language models (LLMs) have emerged as powerful knowledge bases yet are limited by static training data, leading to issues such as hallucinations and safety risks. Editing a model's internal knowledge through the locate-and-edit paradigm has proven a cost-effective alternative to retraining, though current unstructured approaches—especially window-based autoregressive methods—often disrupt the causal dependency between early memory updates and later output tokens. In this work, we first theoretically analyze these limitations and then introduce Matryoshka Unstructured Knowledge Editing ($\mu$KE), a novel memory update mechanism that preserves such dependencies via a Matryoshka-style objective and adaptive loss coefficients. Empirical evaluations on two models across five benchmarks demonstrate that $\mu$KE improves edit efficacy by up to 12.33% over state-of-the-art methods, and remains robust when applied to diverse formatted edits, underscoring its potential for effective unstructured knowledge editing in LLMs.

## 1 Introduction

Large Language Models (LLMs) are increasingly powering a diverse range of applications—from conversational agents (Wang et al., 2024a; Guo et al., 2024) to complex scientific research (OpenAI, 2025). Despite their impressive capabilities, these models are typically trained on static datasets, which can result in issues such as hallucinations (Huang et al., 2025) and other safety risks (Yuan et al., 2024; Xu et al., 2024). Targeting light-weight maintenance of LLMs, model editing (Dai et al., 2022; Mitchell et al., 2022b) has emerged as a cost-effective method for updating a model's internal knowledge without the need for extensive retraining. In particular, the locate-and-edit paradigm (Meng et al., 2022; 2023) provides a promising approach by precisely identifying the internal components where factual information is stored and selectively modifying them to incorporate new, accurate data, all while minimizing collateral changes to unrelated knowledge.

Early locate-and-edit methods (Dai et al., 2022; Meng et al., 2023) primarily focus on editing structured knowledge in the form of (subject, relation, object) triplets, thereby limiting their applicability to more complex, real-world scenarios. Recently, AnyEdit (Jiang et al., 2025) has introduced a chunking strategy that decomposes the task of editing a long target into autoregressively editing a sequence of windows. As illustrated in Figure 1, the editing target answer regarding "the critical temperature change of a superconducting magnet", for example, can be segmented into several sentences as windows. These windows then serve as new targets to be edited one by one, in an autoregressive style, to realize the total effect of editing the whole answer. By this means, AnyEdit extends locate-and-edit algorithms, originally designed for structured editing, to handle unstructured editing tasks.

Despite the significant improvement of AnyEdit to existing locate-and-edit methods such as MEMIT (Meng et al., 2022) and AlphaEdit (Fang et al., 2024) on unstructured knowl-

---

\* Co-first and corresponding authors.

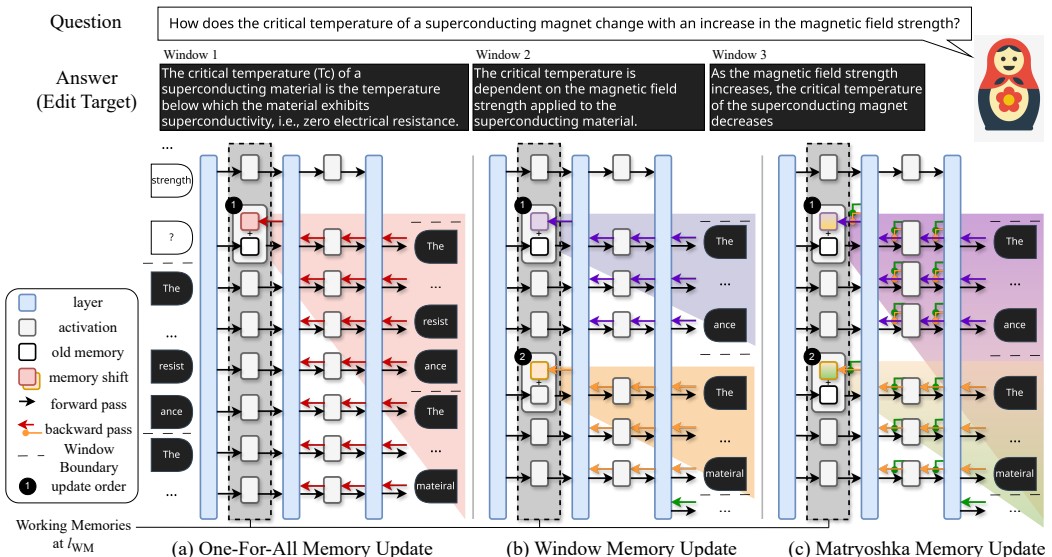

Figure 1: **Comparison between different unstructured editing paradigms in a unified working memory update view.** As shown in the figure, the One-for-All paradigm in (a) updates one memory for the entire long edit target to be edited, being limited by the capacity of one memory shift. The window memory strategy in (b) splits the long target into windows and updates multiple memories autoregressively, however, overlooking memory dependency from former memories to latter target outputs. Our Matryoshka memory update in (c) treats each memory to be potentially contributing to all the target tokens afterwards, maintaining proper dependency while benefiting from the capacity of multiple memory shifts.

edge editing, the window-based autoregressive editing strategy suffers from an intrinsic limitation when combined with the locate-and-edit paradigm. Given that locate-and-edit approaches first compute the updates to the model's internal states at certain located layers (which we denote as *working memories*) and map these updates back to model weights subsequently, a window-based fully autoregressive process implies that there exists no dependency between edited internal states and output tokens at later positions, which is not optimal if we consider a fully retrained model as the relation between internal states and later outputs should be causal. This proposes a general question for unstructured editing that sequentially updates multiple working memories: *how can we properly encode the memory dependency after a sequence of edits on a long target?*

To investigate this question, we first provide a theoretical analysis of the issue regarding missing working memory dependency in AnyEdit. Then, based on the analysis, we present a simple yet effective Matryoshka-style memory update mechanism for unstructured locate-and-edit, which together we call Matryoshka Unstructured Knowledge Editing ($\mu$KE). Our approach conceptualizes the update in an early working memory as a condensed representation shift partially covering all subsequent edit targets, thereby preserving causality between the edited memory and later token distributions. To enforce this dependency and also edit efficacy for each window, $\mu$KE incorporates a Matryoshka-style objective that balances the contribution of one memory update to edits to each range of edit targets. Additionally, we introduce memory update dynamics-informed adaptive coefficients for the loss terms to facilitate better optimization. Comprehensive empirical results on two models and five benchmarks demonstrate that $\mu$KE outperforms AnyEdit, up to 12.33% in edit efficacy, and is more robust to edits of diverse formats, underscoring its effectiveness in unstructured knowledge editing. [1]

---

[1]Our code and data are available at https://github.com/PurCL/muke.

## 2 Preliminaries

**Autoregressive LLM** We consider a transformer architecture LLM (Vaswani et al., 2017) $G$ consisting of $L$ layers to be edited. Given an input $X = [x_1, \cdots, x_T]$, the $l$-th transformer decoder layer forwards internal state at position $t$ as

$$h_t^l = h_t^{l-1} + a_t^l + m_t^l, \ a_t^l = \text{Attn}(h_0^{l-1}, h_1^{l-1}, \cdots, h_t^{l-1}), \ m_t^l = \text{MLP}(h_t^{l-1} + a_t^l). \qquad (1)$$

We omit some components in a transformer, such as the embedding layer and layer normalization to simplify the discussion. The final layer output projected by the LM head and softmax will produce a conditional distribution, denoted as $\mathbb{P}_G(\cdot|X)$. To edit such an LLM is to maximize the conditional probability $\mathbb{P}_G(Y^*|X)$ given a prompt $X$ and edit target $Y^*$ with constraints.

**Locate-and-Edit Directly applied to Unstructured Editing** The locate-and-edit paradigm typically comprises three stages. The first stage leverages causal tracing (Meng et al., 2022) to locate certain layers that are likely to store knowledge. We denote the top located layer as $l_{\text{WM}}$, where the output hidden states are hypothesized to contain the memory of the LLM w.r.t. a (subject, relation) or in general a prompt. In the second stage, one hidden state at a certain position of $l_{\text{WM}}$'s outputs will be considered as the specific working memory that requires updates to enforce editing. To update this outdated memory, a bias term $\delta$ (referred to as "memory shift" in Figure 1) is added to this hidden state by hooking $h_i$, which is an operation supported by popular ML libraries. The transformer execution is then modified from the original $G(\cdots, h_i, \cdots)$ to $G(\cdots, h_i + \delta_i, \cdots)$, producing a new distribution $\mathbb{P}_{G(h_i+\delta)}(\cdot|X)^2$ which allows gradient descent to optimize $\delta$ given the editing target. In the final stage, a batch update process will map the working memory updates to static model weight shifts with constrained optimization (e.g., with locality constraints). For more details, please refer to Appendix C.

As shown in Figure 1.(a), directly applying locate-and-edit to unstructured editing as studied in UnKE (Deng et al., 2024) and AnyEdit (Jiang et al., 2025) is implemented as (1) setting the position of the memory shift $\delta_{\text{global}}$ at the point where the first token of the target is generated (or equivalently the last input token of the prompt); and (2) optimize $\delta_{\text{global}}$ for the target $Y = [y_1, \cdots, y_M]$ by minimizing the following objective,

$$\mathcal{L}_{\text{one}}(\delta_{\text{global}}) = -\frac{1}{M} \sum_{i=1}^{M} \log \mathbb{P}_{G(h_1+\delta_{\text{global}})}(y_i|X, y_{<i}), \qquad (2)$$

which assumes that working memory updates at one position can edit all the later generations. We denote this kind of memory update as the *One-for-All* strategy.

**Window-by-window Unstructured Editing** The bottleneck of this strategy is the limited capacity of working memory, represented by activation at a single time step. AnyEdit (Jiang et al., 2025) proposes to overcome such bottleneck by splitting down the original long edit target into windows $Y_1, \cdots, Y_N$, where $Y = [Y_1; \cdots; Y_N]$. This *Window-by-Window* strategy as shown in Figure 1.(b) leverages multiple working memory updates in a fully autoregressive way w.r.t. windows conditioning on all their previous contexts, i.e., a series of working memories $h_1, \cdots, h_N$ located at the starting time step of each window will be updated via corresponding $\delta_1, \cdots, \delta_N$ as shifts for one long target[3]. The autoregressive process of the model distribution can be described as

$$\mathbb{P}_{G_i(h_1, \cdots, h_i)} \to \mathbb{P}_{G_i(h_1+\delta_1, \cdots, h_i)} \to \cdots \to \mathbb{P}_{G_i(h_1+\delta_1, \cdots, h_i+\delta_i)}, \qquad (3)$$

---

[2]We only highlight core states here for the purpose of differentiating distributions.
[3]Here we abuse the index notation a little bit: the index $i$ of $h_i$ and $\delta_i$ actually maps to the last token position before the $i$-th window in the full context.

where the optimization of each $\delta_i$ has the following objective to minimize negative log likelihood (NLL) of the target window given previous contexts,

$$\mathcal{L}_{\text{win}}(\delta_i) = -\frac{1}{|Y_i|} \sum_{j=1}^{|Y_i|} \log \mathbb{P}_{G(h_1+\delta_1^*,\cdots,h_{i-1}+\delta_{i-1}^*,h_i+\delta_i)}(Y_{i,j}|X,Y_{<i},Y_{i,<j}), \tag{4}$$

where each $\delta^*$ denotes an already optimized memory shift for an early window.

## 3 Matryoshka Unstructured Knowledge Editing

In this section, we first analyze the limitations of the window-by-window memory update strategy (Section 3.1) and then discuss the designs of $\mu$KE (Section 3.2, 3.3).

### 3.1 Window-by-Window Update Overlooks Memory Dependency

The window-by-window strategy overlooks the influence of one memory update on future windows except for the first one. To theoretically understand this limitation, we analyze the difference between two scenarios in updating working memories for one edit target: (1) progressively updating a series of working memories w.r.t. each window, and (2) updating all the working memories in parallel w.r.t. the full target.

For scenario (1), we can derive the gradient of the window-by-window objective from Eq. 4,

$$\nabla \mathcal{L}_{\text{win}}(\delta_i) = -\frac{1}{|Y_i|} \sum_{j=1}^{|Y_i|} \frac{\partial \log \mathbb{P}_{G(h_1+\delta_1^*,\cdots,h_{i-1}+\delta_{i-1}^*,h_i+\delta_i)}(Y_{i,j}|X,Y_{<i},Y_{i,<j})}{\partial \delta_i} \nabla \delta_i. \tag{5}$$

For scenario (2), the objective is simply the NLL of the full target sequence,

$$\mathcal{L}_{\text{prl}}(\delta_1,\cdots,\delta_N) = -\frac{1}{M} \log \mathbb{P}_{G(h_1+\delta_1,\cdots,h_N+\delta_N)}(Y|X) = \frac{1}{M} \sum_{i=1}^{N} -\log \mathbb{P}_{G(h_1+\delta_1,\cdots,h_i+\delta_i)}(Y_i|X,Y_{<i}), \tag{6}$$

and by the chain rule, the gradient of $\mathcal{L}_{\text{prl}}$ can be decomposed into

$$\nabla \mathcal{L}_{\text{prl}}(\delta_1,\cdots,\delta_N) = \frac{1}{M} \sum_{i=1}^{N} \sum_{j=1}^{i} \frac{\partial -\log \mathbb{P}_{G(h_1+\delta_1,\cdots,h_i+\delta_i)}(Y_i|X,Y_{<i})}{\partial \delta_j} \nabla \delta_j \tag{7}$$

$$= \frac{1}{M} \sum_{j=1}^{N} \sum_{i=j}^{N} \frac{\partial -\log \mathbb{P}_{G(h_1+\delta_1,\cdots,h_j+\delta_j)}(Y_i|X,Y_{<i})}{\partial \delta_j} \nabla \delta_j \tag{8}$$

$$= \frac{1}{M} \sum_{j=1}^{N} \left[ |Y_j| \nabla \mathcal{L}_{\text{win}}(\delta_j) + \sum_{i=j+1}^{N} \frac{\partial -\log \mathbb{P}_{G(h_1+\delta_1,\cdots,h_j+\delta_j)}(Y_i|X,Y_{<i})}{\partial \delta_j} \nabla \delta_j \right]. \tag{9}$$

We can see in Eq. 9 that if we optimize $\delta$s together to maximize the likelihood of the entire target, there will be extra gradients, aside from $\nabla \mathcal{L}_{\text{win}}(\delta_i)$, for $\delta_j$ to optimize for all later windows. This indicates the necessity of memory dependency, which is completely ignored in the window-by-window objective.

A naive question would be: *why not directly optimize $\delta$s using $\mathcal{L}_{prl}$?* Empirically, we find that parallel optimization performs less well than sequential one, potentially due to the optimization difficulty with a larger parameter space when combined with the subtle designs like normalization of $\delta$ before each optimization step and the batch update strategy in locate-and-edit algorithms. We add more discussions in Appendix C.3.

Figure 2: **Matryoshka-style working memory update.** For each $\delta_i$, the objective is a weighted sum of negative log-likelihood of all the target figures starting from window $i$ conditioned on previous contexts.

Figure 3: An example of affinities between target figures.

## 3.2 A Matryoshka-Style Working Memory Update Objective

To overcome the limitations of window-by-window strategy, we propose a Matryoshka-style memory update objective $\mathcal{L}_\mu$ for $\mu$KE, enabling gradient flows from all latter tokens to former working memories for optimization when performing sequential updates, as shown in Figure 1.(c). This memory update objective targets a series of condensed representation shifts (partially) covering the corresponding subsequent edit target in the sequential update process. A detailed elaboration of $\mathcal{L}_\mu$ is in Figure 2, which is essentially the mean of weighted NLL loss of all $(Y_i), (Y_i, Y_{i+1}), \cdots, (Y_i, \cdots, Y_N)$ for $\delta_i$, formally expressed as follows,

$$\mathcal{L}_\mu(\delta_i) = -\frac{1}{N-i+1} \Big[ \lambda_{i,i} \log \mathbb{P}_{G(\boldsymbol{h}_1+\delta_1^*, \cdots, \boldsymbol{h}_{i-1}+\delta_{i-1}^*, \boldsymbol{h}_i+\delta_i)}(Y_i|X, Y_{<i}) + \lambda_{i,i+1} \log \mathbb{P}_{G(\cdots, \boldsymbol{h}_i+\delta_i)}(Y_i, Y_{i+1}|X, Y_{<i})$$
$$+ \cdots + \lambda_{i,N} \log \mathbb{P}_{G(\cdots, \boldsymbol{h}_i+\delta_i)}(Y_i, \cdots, Y_N|X, Y_{<i}) \Big]. \quad (10)$$

We call $(Y_i), (Y_i, Y_{i+1}), \cdots, (Y_i, \cdots, Y_N)$ *target figures*, in contrast to windows, which simulate the wooden components of a Matryoshka doll. Intuitively, the objective shows that the edited working memory should contribute to the successful generation of all target figures, instead of biasing towards generating the first window $Y_i$ as in $\mathcal{L}_{\text{win}}$. The $\lambda_{i,j}$ are the adaptive coefficients to balance the importance of target figures in the objective, which will be adjusted per sample. We will discuss it in the next section.

Another way to understand this Matryoshka-style objective, from the perspective of windows, is that the memory update should prioritize former windows, while covering latter ones as much as possible. Despite the dynamic coefficients here, if we reorganize it into the sum of window-wise NLLs,

$$\mathcal{L}_\mu(\delta_i) = -\frac{1}{N-i+1} \sum_{j=i}^{N} \underbrace{\sum_{k=j}^{N} \lambda_{i,k}}_{\text{coefficient for window } j} \log \mathbb{P}_{G(\cdots, \boldsymbol{h}_i+\delta_i)}(Y_j|X, Y_{<j}), \quad (11)$$

we will see that the loss assures that the coefficients of terms w.r.t. early windows are larger than those of later ones, assuming $\lambda_{i,j} > 0$. Given that memories are updated sequentially, there is no second chance for early windows to be edited. Hence, memories need more aggressive optimization towards early windows in the edit target, leaving other parts to later sequential updates. This also explains why we don't directly apply One-for-All sequentially for Matryoshka memory update, as it treats all windows evenly.

## 3.3 Adaptive Coefficients Informed by Affinity Between Edit Target Figures

In practice, $\mathcal{L}_\mu$ has to deal with data samples with different edit difficulty, given fixed learning rates and optimization steps, which is a common practice in existing locate-and-edit algorithms. This has not been a big problem in structured editing, as the object to be edited is typically one token (Meng et al., 2023). However, in the scenario of unstructured

editing where the target length is not constant, the variance between data samples gets larger. Empirically, we find that edit efficacy dramatically varies across different target lengths with a static objective (Figure 5).

Thus, we introduce adaptive coefficients $\lambda_{i,j}$ to balance the contribution of intermediate terms in $\mathcal{L}_\mu$ to mitigate such issues. Inspired by recent studies in data selection via gradient-based data influence in LLM fine-tuning (Xia et al., 2024), we deem that the coefficients of target figures in $\mathcal{L}_\mu$ can leverage information from the direction of gradients of memory updates. In $\mathcal{L}_\mu$, terms regarding target figures share similarity in gradient directions with each other during optimization due to the Matryoshka-style nature (that they overlap). We denote such similarity between target figures as *affinity*, illustrated in Figure 3. As the term of the first figure, which is also the first window, has the highest weight in $\mathcal{L}_\mu$ as demonstrated in Eq. 11, target figures that have better affinity with the first one will benefit more from this gradient direction during optimization. This potentially leads to an overemphasis on these target figures and the neglect of the rest. Hence, we must compensate for the figures that have less affinity for balance. This results in the coefficients negatively correlated with affinity with the first figure for each $\delta_i$, defined as follows,

$$\lambda_{i,j} = 2 - \frac{1}{T_{\text{aff}}} \sum_{t=1}^{T_{\text{aff}}} \left\langle \nabla_{\delta_i^t} \log \mathbb{P}_{G(\cdots, h_i + \delta_i^t)}(Y_i | X, Y_{<i}), \nabla_{\delta_i^t} \log \mathbb{P}_{G(\cdots, h_i + \delta_i^t)}(Y_i, \cdots, Y_j | X, Y_{<i}) \right\rangle, \quad (12)$$

where $T_{\text{aff}}$ is the optimization steps of $\delta_i$ when minimizing NLL of the first figure (window $i$) to obtain a series of $\delta_i^t$s, and $\langle \cdot, \cdot \rangle$ is cosine similarity. Unlike in data selection for LLM pre-training and fine-tuning, where we have to deal with a gigantic parameter space for gradient similarity computation, thanks to the working memory update design in the locate-and-edit paradigm, the gradient space is much smaller, allowing for efficient similarity computation.

## 4 Experiment Setup

In this section, we briefly introduce the setup of our experiments. For more details, please refer to Appendix A.

**Models and Baselines** We conducted experiments on two base LLMs, Qwen2.5-7B-Instruct (Qwen Team, 2024) and Llama3-8B-Instruct (Meta, 2024). For baselines, we compare against original (pre-edited) models, MEMIT (Meng et al., 2023), AlphaEdit (Fang et al., 2024), UnKE (Deng et al., 2024), AnyEdit and AnyEdit*(Jiang et al., 2025).

**Benchmarks** We adopt the benchmarks from AnyEdit to evaluate $\mu$KE, including Un-KEBench (Deng et al., 2024), the AKEW benchmark (Wu et al., 2024), and EditEverything (Jiang et al., 2025), among which AKEW consists of two subsets: AKEW-CounterFact and AKEW-MQuAKE. We add SelfCheckGPT (Manakul et al., 2023) to benchmark hallucination reduction efficacy (Appendix D.1). For locality evaluation, we assess whether knowledge editing preserves the model's general capabilities using two benchmarks: MMLU (Hendrycks et al., 2021a;b) and IFEval (Zhou et al., 2023).

**Metrics** We report BLEU (Papineni et al., 2002), besides ROUGE-L (Lin, 2004) and BERTScore (Zhang* et al., 2020) that have been reported in AnyEdit. These metrics measure the similarity between model's edited outputs and original outputs to assess the effectiveness of editing algorithms. We run three trials for each experiment and report mean values in the main results. Standard deviations are reported in Appendix D.4 due to space limit.

## 5 Results

### 5.1 Long-Form Knowledge Editing Evaluation

To evaluate the long-form knowledge editing ability, we test $\mu$KE along with baseline methods on UnKEBench, AKEW-CounterFact, and AKEW-MQuAKE. Similarly to AnyEdit, we instantiate two versions of $\mu$KE: one based on MEMIT, which we denote as exactly

| LLM | Method | UnKEBench | | | | | | AKEW (CounterFact) | | | | | | AKEW (MQuAKE) | | |
| | | Ori. | | | Para. | | | Ori. | | | Para. | | | Ori. | | |
| | | BLEU | BERT | R-L | BLEU | BERT | R-L | BLEU | BERT | R-L | BLEU | BERT | R-L | BLEU | BERT | R-L |
|---|---|---|---|---|---|---|---|---|---|---|---|---|---|---|---|---|
| Llama3-8B-It | Pre-edited | 20.09 | 71.38 | 25.49 | 19.80 | 71.29 | 25.31 | 13.89 | 68.33 | 18.76 | 15.81 | 42.22 | 13.18 | 18.84 | 69.79 | 21.16 |
| | MEMIT | 24.97 | 76.21 | 30.03 | 22.51 | 74.50 | 28.29 | 31.88 | 75.83 | 31.23 | 17.52 | 47.27 | 15.88 | 26.61 | 69.93 | 25.75 |
| | AlphaEdit | 21.43 | 73.76 | 26.65 | 20.38 | 72.85 | 25.88 | 23.52 | 72.54 | 24.98 | 16.39 | 45.13 | 14.07 | 22.57 | 69.78 | 22.74 |
| | AnyEdit | 77.91 | 95.69 | 90.36 | 66.29 | 92.38 | 80.51 | 86.34 | 97.86 | 95.51 | 39.48 | 65.31 | 46.26 | 87.28 | 97.38 | 94.28 |
| | μKE | 90.24 | 97.70 | 91.75 | 76.72 | 93.78 | 77.59 | 94.72 | 98.75 | 93.76 | 41.22 | 63.72 | 37.40 | 90.53 | 97.38 | 89.79 |
| | UnKE | 93.20 | 98.09 | 92.94 | 78.77 | 93.33 | 78.94 | 98.26 | 99.61 | 98.25 | 38.63 | 60.28 | 34.40 | 95.04 | 98.93 | 94.40 |
| | AnyEdit* | 99.54 | 99.79 | 99.62 | 72.96 | 91.63 | 77.48 | 99.93 | 99.99 | 99.96 | 40.25 | 62.95 | 42.71 | 99.79 | 99.97 | 99.87 |
| | μKE* | 99.81 | 99.97 | 99.79 | 77.82 | 93.75 | 80.08 | 99.96 | 99.99 | 99.96 | 43.60 | 64.53 | 40.93 | 99.91 | 99.99 | 99.94 |
| Qwen2.5-7B-It | Pre-edited | 20.92 | 72.58 | 27.64 | 20.51 | 72.18 | 26.60 | 28.15 | 69.66 | 20.98 | 29.54 | 52.15 | 19.31 | 24.12 | 69.05 | 21.03 |
| | MEMIT | 45.18 | 78.10 | 38.21 | 40.89 | 76.63 | 34.19 | 45.08 | 77.16 | 38.95 | 32.99 | 56.15 | 25.80 | 41.54 | 71.61 | 35.46 |
| | AlphaEdit | 49.75 | 80.57 | 42.93 | 45.37 | 78.41 | 38.42 | 49.88 | 80.42 | 45.35 | 34.65 | 56.94 | 27.74 | 45.14 | 72.76 | 39.83 |
| | AnyEdit | 86.86 | 96.85 | 91.68 | 64.94 | 90.11 | 71.47 | 88.52 | 97.29 | 91.94 | 38.10 | 62.72 | 39.43 | 86.69 | 96.48 | 90.57 |
| | μKE | 96.34 | 99.02 | 97.02 | 75.44 | 92.25 | 75.34 | 95.23 | 98.76 | 95.57 | 45.20 | 64.89 | 39.32 | 94.75 | 98.42 | 95.45 |
| | UnKE | 91.72 | 96.85 | 90.79 | 56.79 | 83.85 | 51.97 | 91.87 | 97.54 | 91.11 | 38.45 | 59.15 | 29.16 | 88.04 | 95.15 | 86.31 |
| | AnyEdit* | 96.13 | 98.85 | 97.30 | 47.17 | 80.75 | 50.96 | 96.70 | 99.10 | 97.48 | 32.02 | 57.29 | 31.94 | 97.44 | 99.49 | 97.70 |
| | μKE* | 99.27 | 99.61 | 99.24 | 50.69 | 81.57 | 48.33 | 99.68 | 99.89 | 99.67 | 35.20 | 57.46 | 29.41 | 99.38 | 99.74 | 99.33 |

Table 1: **Performance of different methods on long-form knowledge editing.** "Pre-edited" refers to the original pre-trained LLM. Best results within structured-derived or unstructured-derived methods are highlighted in **blue and bold**; the overall best result across both groups is highlighted with a **blue background**. "Ori." denotes the performance of the edited model on the original questions and is analogous to edit success used in earlier works. "Para." measures performance on paraphrased versions of the original questions, reflecting the generalizability of the edited knowledge.

"μKE", and the other based on UnKE, denoted as "μKE*". This allows us to directly compare with AnyEdit and AnyEdit*, which are also based on these two locate-and-edit algorithms. Unless otherwise specified, we set the edit batch size to 1 and the decoding temperature to 0.001, following AnyEdit.

As shown in Table 1, μKE demonstrates significant improvements over AnyEdit in BLEU scores, up to 12.33% with the original question and 10.5% with the paraphrased question of absolute improvement, while producing higher or competitive ROUGE-L scores in most cases. This demonstrates the effectiveness of a Matryoshka-style objective that models memory dependency in unstructured editing. μKE* further amplifies this performance, achieving near-perfect scores in BLEU, ROUGE-L, and BERTScore, reaching values as high as 99.996%, bringing the editing success in close proximity to 100%.

We observe that there are cases where MEMIT-based methods surpass UnKE-based methods in results regarding paraphrased questions, which indicates better generalizability, especially when editing Qwen2.5-7B-Instruct. We believe this can be attributed to the difference in total editable parameters, in addition to base LLM and layer localization differences, which are also fundamental. MEMIT-based methods rely on the core assumption that transformer feedforward layers are key value memories (Geva et al., 2021) and only update the down projection layer. On the other hand, UnKE-based methods allow updates in full transformer layers using gradient descent, potentially leading to the overfitting (Zhang et al., 2024). Overall, the results indicate that either μKE or μKE* consistently achieves the strongest generalization performance across the majority of evaluation settings. For example, μKE has significantly better results for paraphrased questions than UnKE on UnKEBench with Qwen2.5-7B-Instruct (+18.65% BLEU, +8.50% BERTScore, and +23.37% Rouge-L). Meanwhile, it also performs better than the MEMIT-based AnyEdit counterpart.

## 5.2 Diverse-Formatted Knowledge Editing Evaluation

To further evaluate the effectiveness of our proposed method for knowledge editing in diverse domains and problem structures, we test it on EditEverything, which encompasses a

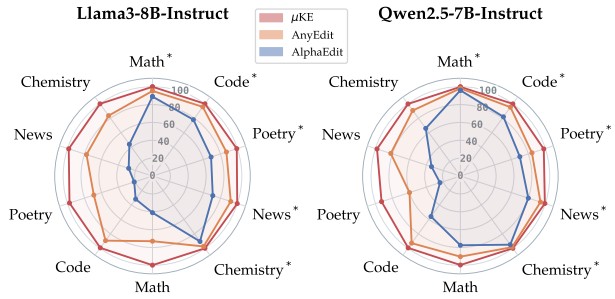
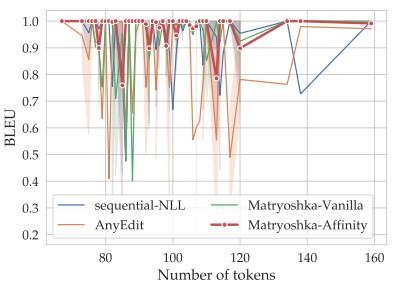

Figure 4: Performance comparison between $\mu$KE and baseline methods on long-form, diverse-formatted knowledge. Knowledge types without * represent the metric Rouge-L, while those with * indicate BERTScore.

Figure 5: BLEU of answers generated by edited model varying in original answer length.

wide range of domains, including mathematics, poetry, news, programming, and chemistry. Figure 4 shows the edit success of $\mu$KE and baseline methods. $\mu$KE outperforms AnyEdit across all sub-domains of the EditEverything dataset, demonstrating the effectiveness of $\mu$KE in handling diverse-formatted editing tasks. Moreover, $\mu$KE exhibits robust and consistent performance across different domains, whereas AnyEdit shows a noticeable degradation on the Poetry subset, reflected by a significant drop in ROUGE-L scores on both evaluated LLMs. We include an example demonstrating the effectiveness of $\mu$KE over AnyEdit on the poetry subset of the EditEverything dataset in Appendix D.7.1.

## 5.3 General Capability Preservation Evaluation

How a model preserves its general capability after editing, i.e., the "locality" of edited models, has been a core aspect of evaluating editing algorithms. To assess whether an edited LLM preserves general functionality, we focus on two core capabilities of instruction-tuned models: natural language understanding (MMLU) and instruction following (IFEval).

As shown in Table 2, both $\mu$KE and AnyEdit generally maintain stable performance in preserving original model capabilities while integrating new knowledge. We notice some model-wise difference as the performance of AnyEdit drops a bit on IFEval with Llama3-8B-It and similarly $\mu$KE with Qwen2.5-7B-It. We think this issue is partly due to the granularity of localization. Note that AnyEdit and $\mu$KE inherit predefined editing layer selection from MEMIT and UnKE, where we consider all working memory shifts happen on the same layer(s). Intuitively, an unstructured editing target, which can be long, can encompass knowledge of different levels, which likely requires more fine-grained designs in localization and corresponding updates. This further echoes the importance of memory dependency through multiple layers for unstructured knowledge editing.

| LLM | Method | MMLU | IFEval | |
| | | acc | strict | loose |
|---|---|---|---|---|
| Llama3-8B-It | Pre-edited | 67.16±0.38 | 69.50±1.98 | 75.60±1.85 |
| | AnyEdit | 65.81±0.12 | 64.21±0.12 | 64.21±0.65 |
| | $\mu$KE | 65.59±0.12 | 63.01±0.66 | 71.56±0.61 |
| | AnyEdit* | 65.72±0.12 | 70.13±0.62 | 75.91±0.58 |
| | $\mu$KE* | 64.99±0.12 | 69.13±0.63 | 75.15±0.59 |
| Qwen2.5-7B-It | Pre-edited | 73.45±0.36 | 70.24±1.97 | 73.57±1.90 |
| | AnyEdit | 72.21±0.11 | 65.56±0.65 | 68.74±0.63 |
| | $\mu$KE | 71.64±0.11 | 60.87±0.66 | 64.15±0.65 |
| | AnyEdit* | 73.28±0.11 | 70.89±0.62 | 73.71±0.60 |
| | $\mu$KE* | 73.25±0.11 | 70.63±0.62 | 73.61±0.60 |

Table 2: Evaluation results on MMLU and IFEval for pre-edited and post-edited LLMs.

## 6 Analysis

We conduct further analysis to justify some design choices in $\mu$KE.

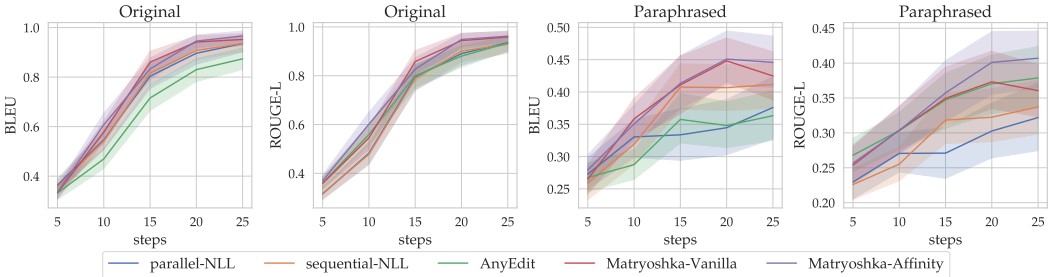

Figure 6: Comparison of memory update optimization stability between different methods.

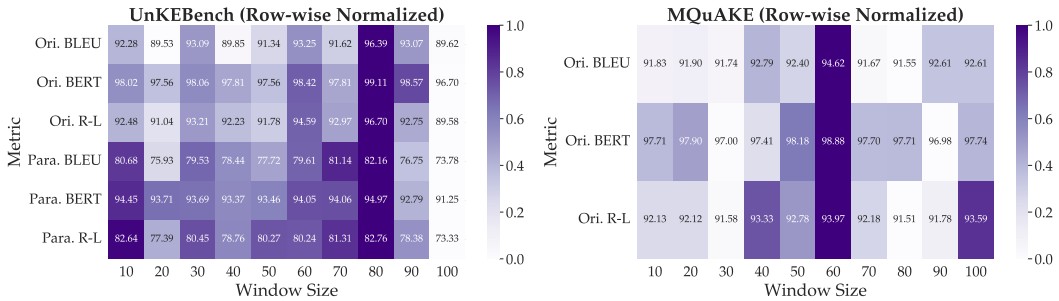

Figure 7: $\mu$KE performance with various window sizes.

**Performance of $\mu$KE is less affected by edit target length.** We compare the performance of different memory update methods regarding different data samples of various lengths. Here, "parallel-NLL" denotes the simultaneous updates of multiple working memories with an NLL loss as mentioned in Section 3.1; "sequential-NLL" denotes applying One-for-All (NLL starting from the corresponding window of the memory to the end of the target) to sequential memory update; "Matryoshka-Vanilla" denotes $\mathcal{L}_\mu$ with constant coefficients $\lambda_{i,j} = 1$; and "Matryoshka-Affinity" denotes $\mathcal{L}_\mu$. We adopt identical window size (20), learning rate (0.5), and number of optimization steps (25) for all methods for fair comparison. We use only 100 data samples for this study. As shown in Figure 5, we observe that the BLEU scores of answers generated by edited models typically vary greatly for static objectives such as AnyEdit's window-based objective or "Matryoshka-Vanilla". The adaptive $\mathcal{L}_\mu$ has shown the best robustness to target length.

**$\mathcal{L}_\mu$ has better optimization stability.** Under the same setting as described above, we compare the optimization stability of different methods. We evaluate each method when optimized with steps ranging from 5 to 25 and observe the performance change. As shown in Figure 6, we can see that (1) parallel optimization of multiple memory updates is harder than sequential due to increased parameter space; (2) both Matryoshka-style methods perform better than sequential-NLL, indicating the significance of weighting in objective design; (3) $\mathcal{L}_\mu$ with dynamic coefficients is more stable than with static ones (smoothly increases), especially when we refer to results regarding paraphrased questions.

**Impact of window size on $\mu$KE performance.** As shown in Figure 7, we conduct an ablation study to evaluate the impact of window size on the performance of $\mu$KE. Keeping all other settings identical to those described above, we apply $\mu$KE to edit Llama3-8B-Instruct using window sizes ranging from 10 to 100 tokens. The results suggest that $\mu$KE 's performance is relatively insensitive to window size, with variations across all metrics typically within 3%. Importantly, our reported results do not rely on tuning for an optimal window size; instead, we select the window size used in our main experiments based on a few initial trials. Interestingly, we observe dataset-wise bias: for UnKEBench, a window size of 80 yields the best results across all three metrics, whereas for MQuAKE, a window

size of 60 performs best. This observation suggests the potential for further improvement by dynamically selecting the window size based on the characteristics of the input data, which we leave as a promising direction for future work.

# 7 Related Work

**Unstructured Editing of LLMs**   Early research in model editing focused on structured factual updates, positing that transformer layers store discrete key–value pairs where editing a specific neuron or a set of neurons (Geva et al., 2021; Meng et al., 2022; 2023) could correct erroneous outputs. However, this perspective does not fully capture the intricacies of unstructured, free-form knowledge. Recently, new benchmarks and methods such as DUnE (Akyürek et al., 2023), AKEW (Wu et al., 2024), UnKE (Deng et al., 2024), and AnyEdit (Jiang et al., 2025) have emerged. These studies extend the editing paradigm to handle complex, longer-form, and noisier textual content that traditional structured methods cannot easily address.

**Model Editing Methods**   Model editing strategies have been explored in width and depth. The locate-then-edit framework (Dai et al., 2022; Meng et al., 2022), which targets and modifies specific neurons, weight matrices or layers responsible for factual inaccuracies, has been refined by methods like RECT (Gu et al., 2024) and AlphaEdit (Fang et al., 2024). Retrieval-based methods (Mitchell et al., 2022b; Zheng et al., 2023) and meta-learning approaches (Mitchell et al., 2022a; Tan et al., 2024) offer alternatives that adjust outputs without directly altering internal parameters. Further advancements in lifelong or continual editing—using mechanisms such as adapter modules (Hartvigsen et al., 2024; Wang et al., 2024b) and minimal weight updates (Sutton et al., 2024)—aim to support sequential and scalable updates. There are also investigations into multi-hop editing (Zhong et al., 2023), and task arithmetics (Ortiz-Jimenez et al., 2024; Ilharco et al., 2023).

# 8 Conclusion

In this work, we addressed key limitations in unstructured knowledge editing by analyzing the disruption of causal memory dependencies inherent in existing window-based autoregressive methods. We introduced Matryoshka Unstructured Knowledge Editing ($\mu$KE), a novel framework that leverages a Matryoshka-style memory update mechanism and adaptive loss coefficients to preserve the critical dependency between early memory updates and subsequent output tokens. Comprehensive empirical evaluations on two models across five benchmarks demonstrate that $\mu$KE not only enhances editing efficacy compared to state-of-the-art techniques but also provides a robust approach to updating large language models.

**Limitations & Future Work**   The locate-and-edit paradigm for unstructured editing is tightly coupled with the optimization process embedded in its workflow. Although $\mu$KE alleviates this issue through a sample-wise adaptive loss design, further exploration into more effective and generalizable optimization strategies remains an important direction for future work.

Additionally, our evaluation relies on existing benchmarks for unstructured knowledge editing. While these benchmarks cover diverse formats and target lengths, they lack explicit annotations of fine-grained knowledge units within longer targets. As a result, they provide limited insight into the actual efficacy of edits and other nuanced evaluation dimensions. Future research should aim to design more comprehensive benchmarks and metrics that capture a broader range of editing behaviors—such as editing multiple interdependent knowledge pieces while maintaining consistency with causal memory dependencies.

Finally, knowledge units within long edit targets are not necessarily uniform in length like fixed-size windows. A promising avenue for future work involves dynamically identifying and localizing knowledge units within raw input sequences and targets. Achieving this would likely require developing a more intrinsic and fine-grained model of memory dependency and causality in autoregressive language models.

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

# A    Experiment Details

## A.1    Evaluation Details

**Locality**    To test general ability preservation, we evaluated edited models on the MMLU benchmark (Hendrycks et al., 2021a;b) and the IFEval benchmark (Zhou et al., 2023). MMLU is a multiple-choice test that covers a wide range of subjects to test general understanding. IFEval evaluates responses to instruction-following prompts, measuring the fulfillment of a wide range of natural language requests.

For MMLU, we use 5-shot prompts formatted as multi-turn conversations, with a batch size of 8. For IFEval, we ran the edited models in a 0-shot setting with a batch size of 16, and computed prompt-level accuracy under two settings: strict, which requires exact matches to reference answers, and loose, which accepts semantically equivalent responses.

All experiments were conducted on an AWS EC2 p4d.24xlarge instance using four NVIDIA A100 SXM4 GPUs (40GB each). Models were loaded in bfloat16. Chat templates were applied during tokenization. For Qwen2.5-7B-Instruct models, we used the system prompt: "You are Qwen, created by Alibaba Cloud. You are a helpful assistant." For Llama3-8B-Instruct models, we used: "You are a helpful assistant." We repeated each experiment ten times, and we report the mean and standard error, reweighted using inverse-variance weighting.

**Reproducing the baseline**    We followed the experimental settings of AnyEdit and made our best effort to replicate the baseline results. The key configurations are as follows: for the MEMIT-based method, we edited layers 4 to 8 of the down-projection and used chunk sizes of 40 and 50 tokens for the LLaMA 3 and Qwen 2.5 models, respectively, with no overlap. Each chunk was optimized for up to 25 steps. For the UnKE-based method, we edited the 7th layer of both models using the same chunk sizes, with 50 optimization steps. Updates to key-value representations were limited to a maximum of 25 steps. We conducted all experiments on an AWS EC2 p5e.48xlarge instance, using the float32 data type. We repeated each experiment three times and report the mean.

## A.2    Implementation Details

For computing $\lambda_{i,j}$, we set $T_{\text{aff}}$ to 3 with a learning rate of 0.5. If not otherwise specified, we optimize $\mu$KE for 25 steps, which is consistent with baselines for fairness.

Similarly to AnyEdit and AnyEdit$^*$, our $\mu$KE and $\mu$KE$^*$ are built on top of MEMIT and UnKE. For Llama3-8B-Instruct, we set the window size to 30 tokens without overlap, and for Qwen2.5-7B-Instruct, we set the window size to 20 tokens without overlap.

We use the same computing environment as the reproduction of baseline methods.

# B    Broader Definition of "Working Memory"

In this paper, we denote "working memories" as the specific hidden states corresponding to the located layers that are hypothesized to contain the knowledge to be edited and can be updated with a bias term using gradient descent. Conceptually, this can be connected to the definition in human working memory in psychology (Hitch & Baddeley, 1976) and cognitive neuroscience (Finn et al., 2019; Degutis et al., 2024). In short, both in human cognition and in the locate-and-edit algorithms, "working memory" acts as a short-lived, capacity-limited store that is actively manipulated (through attention or gradient steps) to change future outcomes (behavioral response or token probabilities). The cognitive-science emphasis on neural circuits in prefrontal/parietal areas echoes the $\mu$KE reliance on a specific transformer layer ($l_{\text{WM}}$) whose activations effectively serve the same functional purpose: keeping relevant information "online" so that it can be selectively updated and then used to guide subsequent processing.

# C Locate-and-Edit Paradigm Details

We introduce details of the locate-and-edit paradigm (Meng et al., 2022) in this section, which covers representative locate-and-edit algorithms: MEMIT (Meng et al., 2023), AlphaEdit (Fang et al., 2024), and UnKE (Deng et al., 2024).

## C.1 Summary of Symbols

We summarize a comprehensive list of all notations in Table 3.

| Notation | Meaning |
|----------|---------|
| $X$ | Input containing token $x_1, \cdots, x_T$. |
| $Y$ | Output containing token $y_1, \cdots, y_M$. |
| $Y_i$ | The $i$-th window of $Y$. $Y = [Y_1, \cdots, Y_N]$. |
| $\|Y_i\|$ | Number of tokens in $Y_i$. |
| $\mathbb{P}$ | A probabilistic distribution. |
| $\mathbb{P}(Y\|X)$ | The conditional distribution of output $Y$ given $X$. |
| $\mathbb{P}(y_i\|X, y_{<i})$ | The conditional distribution of the $i$-th token $y_i$ given input $X$ and previous $y_1, \cdots, y_{i-1}$. |
| $\mathbb{P}(Y_i\|X, Y_{<i})$ | The conditional probability of window $Y_i$ given input $X$ and previous windows $Y_1, \cdots, Y_{i-1}$. It is essentially the product of the conditional probability of each token within the window. |
| $G$ | A transformer model. |
| $l_{\text{WM}}$ | Layer of the working memory to be updated to enforce edits. |
| $h_i^l$ | A hidden state / activation of transformer at layer $l$ and position $i$. |
| $\delta_i^l$ | Corresponding shift to $h_i^l$ to realize edit in the output. |
| $G(h_i^l + \delta_i^l)$ | A transformer that has been hooked at the hidden state at layer $l$ and position $i$. The value of the original $h_i^l$ is changed to $h_i^l + \delta_i^l$ during the forward process. This will affect all later hidden states and the final output distribution. |
| $\mathbb{P}_{G(h_i + \delta_i)}$ | The output distribution of a hooked transformer. Note that we abuse the index $i$ here, and it in particular indicates the index of the last token before the $i$-th window. |
| $\nabla_\delta \mathcal{L}, \nabla \mathcal{L}(\delta)$ | Gradient of $\delta$ w.r.t. $\mathcal{L}$. |
| $T_{\text{aff}}$ | Optimization steps for computation affinity. |
| $\langle \cdot, \cdot \rangle$ | Cosine similarity. |

Table 3: Glossary of Notations.

## C.2 Key-Value Memory in Transformer

Locate-and-edit typically assumes MLP layers are linear associative memory (Geva et al., 2021; Meng et al., 2022) of factual knowledge stored in transformers. Such neural memory mechanism (Sukhbaatar et al., 2015) can be described as

$$m_t^l = \underbrace{W_{\text{down}}^t}_{\text{values}} \cdot \underbrace{\sigma(W_{\text{up}}^t \gamma(h_t^{l-1} + a^l))}_{\text{key}} \tag{13}$$

where $\gamma$ is layer normalization. Hence, only $W_{\text{down}}$, which represents all the memory of a certain layer, needs to be updated to enforce some edits.

UnKE (Deng et al., 2024) extends the idea of key-value memory to multiple transformer decoder layers instead of a local MLP layer, which can be described as,

$$K = G^{<l}(X), \; V(\cdot) = G^l(\cdot), \; m_t^l = V(K)[t] \tag{14}$$

where the key if hidden states of the layer before located $l_{\text{WM}}$, and the full layer $l_{\text{WM}}$ serves as a non-linear value projection to map the key to working memory. In this setup, the weights to be updated is all the parameters within layer $l_{\text{WM}}$.

## C.3 Updating Working Memories

As discussed in the main text, a specific layer $l_{\text{WM}}$ will be located by causal tracing (Meng et al., 2022), and the output hidden states of it will be considered as the working memories related to the edit. Even though there is some difference between the key-value memory of UnKE and others, the working memory update process is identical. By hooking the hidden state $h_i$ at the specific position $i$ to add a memory shift $\delta_i$, we can enforce the edit by performing gradient descent on $\delta_i$, with a fixed number of steps and learning rate universal to all edits, as there is no further information about the optimization as in existing locate-and-edit algorithms.

Aside from the main objectives, during the multi-step optimization using an Adam optimizer (Kingma & Ba, 2014), a detailed per-step normalization of the current delta will be applied. The normalization step projects the new $\delta_i^t$ after the optimization step within the L2 ball of the hidden state $h_i^t$ to be hooked, which can be expressed as

$$\tilde{\delta}_i^t = \begin{cases} \delta_i^t, & \|\delta_i^t\| \leq c \cdot \|h_i\|, \\ \frac{c \cdot \|h_i\|}{\|\delta_i\|} \delta_i, & \|\delta_i^t\| > c \cdot \|h_i\|, \end{cases} \tag{15}$$

where $c$ is the clamping factor. Such projection is intended for preserving locality. Empirically, we found that removing such normalization will improve edit success while harming locality. Such normalization poses extra difficulty to optimization as it changes the dynamics of Adam. Moreover, things become trickier if we include multiple $\delta$s for parallel optimization as discussed in the main text, which we hypothesize would be the underlying reason why we need sequential updates of a series of working memories instead of optimizing them simultaneously.

## C.4 Batch Update

The goal of the batch update stage is to map updates in working memories, i.e., the optimized $\delta_i$s, to actual weight shifts in models.

For locate-and-edit algorithms that assume $W_{\text{down}}$ as the weight to be updated, such mapping is achieved by solving a constrained least squares problem. Define a memory modifying set $\{K_1, M_1\}$ and a memory preserving set $\{K_0, M_0\}$ as follows,

$$K_0 = [k_1|k_2|\cdots|k_n], M_0 = [m_1|m_2|\cdots|m_n],$$
$$K_1 = [k_{n+1}|k_{n+2}|\cdots|k_{n+u}], M_1 = [m_{n+1}^*|m_{n+2}^*|\cdots|m_{n+u}^*], \tag{16}$$

where the $k$ is the key part defined in Eq. 13, and the $m$ is the original working memory, and the $m^*$ is the updated working memory. In practice, the preservation set is created from some random samples that are irrelevant to the editing samples. For unstructured editing, the updated working memories of different windows/figures are considered as separate $m^*$s in this batch update formulation.

The objective function of MEMIT is,

$$W^* = \underset{W}{\arg\min} \left( \sum_{i=1}^{n} \|Wk_i - m_i\|_2^2 + \sum_{i=n+1}^{n+u} \|Wk_i - m_i\|_2^2 \right), \tag{17}$$

where we denote the original weight matrix as $W_0$ here. The closed-form solution to this objective is,

$$W^* = W_0 + (M_1 - W_0 K_1) K_1^\top (K_0 K_0^\top + K_1 K_1^\top)^{-1}. \tag{18}$$

AlphaEdit improved over MEMIT's objective by further constraining that the weight update of $W_{\text{down}}$ is always projected onto the null space of $K_0 K_0^T$, resulting in the following solution,

$$W^* = W_0 + (M_1 - W_0 K_1) K_1^\top P (K_p K_p^\top P + K_1 K_1^\top P + I)^{-1}, \tag{19}$$

where $P$ is a null space projection matrix (Wang et al., 2021).

For UnKE, as the weight update is not limited to one matrix, gradient descent is leveraged to optimize the weight of one transformer decoder layer with the following objective,

$$\theta_l^* = \underset{\theta_l}{\operatorname{argmin}} \left( \sum_{i=1}^{n} \left\| G_{\theta_l}^l(k_i) - m_i \right\|_2^2 + \sum_{i=n+1}^{n+u} \left\| G_{\theta_l}^l(k_i) - m_i^* \right\|_2^2 \right). \tag{20}$$

## D  Additional Results

### D.1  Unstructured Editing for Hallucination Reduction

|  | BLEU | BERTScore | Rouge-L |
|---|---|---|---|
| Pre-Edit | 10.58 | 42.13 | 21.93 |
| AnyEdit | 66.59 | 82.64 | 84.66 |
| $\mu$KE | **86.77** | **91.80** | **87.17** |

Table 4: SelfCheckGPT results for hallucination reduction.

To understand the effectiveness of the proposed $\mu$KE under a more realistic scenario. We conduct additional experiments on SelfCheckGPT (Manakul et al., 2023). We report results from pre-edit model, AnyEdit, and $\mu$KE based on Llama3-8B-Instruct in Table 4. $\mu$KE consistently shows better performance than AnyEdit and significantly reduces hallucinations compared to the pre-edit model.

### D.2  Comparison with Lifelong Editing Baselines

Some of the existing lifelong editing algorithms that leverage additional modules with routing mechanism naturally apply to unstructured editing. To compare them, we experiment with a recent lifelong editing approach WISE (Wang et al., 2024b). We adopt the implementation and hyperparameters of WISE from EasyEdit [4]. In Table 5, 6, 7, and 8, we report the results for Llama3-8B-Instruct and Qwen2.5-7B-Instruct on four benchmarks. We observe incomparable and unstable performance of WISE compared to AnyEdit and $\mu$KE under the default settings.

### D.3  Batch Editing

To understand the effectiveness of $\mu$KE for batch editing, we follow the common practice in locate-and-edit paradigm as described in section C.4 to adapt both AnyEdit and $\mu$KE to batch updates, where the core difference between structured editing and unstructured editing with a batch of edits is that we treats all the $(h_i^l + \delta_i^l)$s for each target in a batch as isolated updated memories (memory modifying set) and combine them into the $M_1$ in Eq. 16. At first glance, this approach seems to introduce emphasis on longer targets with more windows. We therefore investigated a length-normalized variant. However, empirical results indicate that

---

[4] https://github.com/zjunlp/EasyEdit

| Model | Ori. BLEU | Ori. BERT | Ori. R-L | Para. BLEU | Para. BERT | Para. R-L |
|-------|-----------|-----------|----------|------------|------------|-----------|
| Llama | 17.14 | 77.90 | 62.75 | 16.18 | 74.64 | 55.35 |
| Qwen | 24.64 | 78.15 | 55.42 | 25.70 | 77.29 | 52.81 |

Table 5: Results of WISE on UnKEBench.

| Model | Ori. BLEU | Ori. BERT | Ori. R-L | Para. BLEU | Para. BERT | Para. R-L |
|-------|-----------|-----------|----------|------------|------------|-----------|
| Llama | 18.04 | 86.79 | 93.76 | 13.73 | 66.21 | 53.95 |
| Qwen | 32.80 | 73.23 | 31.37 | 30.96 | 54.38 | 25.88 |

Table 6: Results of WISE on CounterFact.

| Model | Ori. BLEU | Ori. BERT | Ori. R-L |
|-------|-----------|-----------|----------|
| Llama | 20.76 | 87.26 | 94.03 |
| Qwen | 35.46 | 77.30 | 46.16 |

Table 7: Results of WISE on MQuAKE.

| Model | Math | | Code | | Poetry | | News | | Chemistry | |
|-------|------|------|------|------|--------|------|------|------|-----------|------|
| | BERT | R-L | BERT | R-L | BERT | R-L | BERT | R-L | BERT | R-L |
| Llama | 90.39 | 61.58 | 84.92 | 87.12 | 90.19 | 91.96 | 87.83 | 75.89 | 97.25 | 92.15 |
| Qwen | 94.38 | 72.73 | 77.34 | 70.61 | 70.51 | 41.87 | 74.55 | 40.49 | 92.76 | 65.51 |

Table 8: Results of WISE on EditEverything.

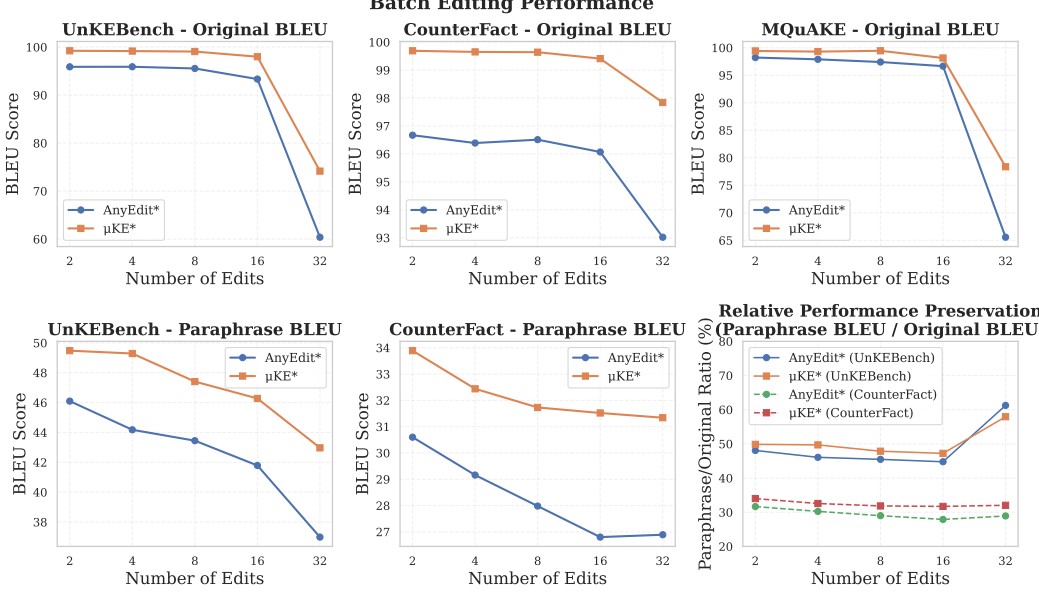

Figure 8: Batch editing performance of $\mu$KE and AnyEdit on three benchmarks.

such normalization slows convergence and leads to underfitting within a fixed optimization budget. Hence, we adopt the straightforward solution for our experiments.

We report results of $\mu$KE and AnyEdit w.r.t different batch sizes in Figure 8. $\mu$KE consistently outperforms AnyEdit for different batch sizes, highlighting the significance of maintaining memory dependency for unstructured batch editing.

| Dataset / Model | Ori. BLEU | Ori. BERT | Ori. R-L | Para. BLEU | Para. BERT | Para. R-L |
|---|---|---|---|---|---|---|
| **UnKEBench** | | | | | | |
| Llama, $\mu$KE | 90.24±0.16 | 97.70±0.04 | 91.75±0.07 | 76.72±0.08 | 93.78±0.05 | 77.59±0.13 |
| Llama, $\mu$KE* | 99.81±0.05 | 99.97±0.02 | 99.79±0.05 | 77.82±0.07 | 93.75±0.03 | 80.08±0.00 |
| Qwen, $\mu$KE | 96.34±0.24 | 99.02±0.06 | 97.02±0.11 | 75.44±0.29 | 92.25±0.06 | 75.34±0.36 |
| Qwen, $\mu$KE* | 99.27±0.05 | 99.61±0.01 | 99.24±0.05 | 50.69±0.54 | 81.57±0.60 | 48.33±0.01 |
| **AKEW-CounterFact** | | | | | | |
| Llama, $\mu$KE | 94.72±0.16 | 98.75±0.03 | 93.76±0.89 | 41.22±0.26 | 63.72±0.14 | 37.40±0.12 |
| Llama, $\mu$KE* | 99.96±0.02 | 99.99±0.00 | 99.96±0.03 | 43.60±0.42 | 64.53±0.40 | 40.93±0.16 |
| Qwen, $\mu$KE | 95.23±0.03 | 98.76±0.09 | 95.57±0.18 | 45.20±0.16 | 64.89±0.16 | 39.32±0.09 |
| Qwen, $\mu$KE* | 99.68±0.04 | 99.89±0.02 | 99.67±0.06 | 35.20±0.90 | 57.46±0.52 | 29.41±0.48 |
| **AKEW-MQuAKE** | | | | | | |
| Llama, $\mu$KE | 90.53±0.21 | 97.38±0.16 | 89.79±0.29 | – | – | – |
| Llama, $\mu$KE* | 99.91±0.04 | 99.99±0.00 | 99.94±0.03 | – | – | – |
| Qwen, $\mu$KE | 94.75±0.12 | 98.42±0.20 | 95.45±0.12 | – | – | – |
| Qwen, $\mu$KE* | 99.38±0.09 | 99.74±0.01 | 99.33±0.06 | – | – | – |

Table 9: Performance of $\mu$KE and $\mu$KE* on UnKEBench, CounterFact, and AKEW-MQuAKE. "–" indicates that paraphrased metrics are not applicable for AKEW-MQuAKE.

| | Ori. BLEU | Ori. BERT | Ori. ROUGE-L | Para. BLEU | Para. BERT | Para. ROUGE-L |
|---|---|---|---|---|---|---|
| matryoshka-vanilla | 96.25 | 98.70 | 96.83 | 41.91 | 63.10 | 37.56 |
| adaptive, step=1 | **96.98** | 99.49 | 97.00 | 44.83 | 65.73 | 40.95 |
| adaptive, step=2 | 96.68 | **99.56** | **97.04** | 45.16 | 65.93 | 41.00 |
| adaptive, step=3 | 96.03 | 99.19 | 96.21 | **45.66** | **66.82** | **41.50** |

Table 10: Ablation study on the number of steps for adaptive coefficients.

## D.4 Statistical Reliability

We report the complete results of $\mu$KE and $\mu$KE$^*$, including the mean of three runs and the standard deviations in Table 9.

## D.5 Ablation Study on Adaptive Coefficients

We conduct an ablation study on the selection of coefficients to further understand their influence. We experiment with Matryoshka-Vanilla (with all coefficients set to 1), and Matryoshka-Affinity with coefficients computed with different steps of optimization. Results are based on Qwen2.5-7B-Instruct, 100 samples in Table 10. We can see that Matryoshka-Vanilla and that of adaptive coefficients are effective in edit success, yet adaptive coefficients help in improving generality of the edit (paraphrased question results), which is consistent with Figure 6.

## D.6 Computational Overhead

As $\mu$KE introduces additional computation cost, we report computation overhead in Table 11. We see a 4.2% increase in time elapsed running with cache and 33.8% without cache, which includes IO costs and intermediate inference, among others. We observe a negligible peak VRAM usage increase. Since efficiency is not the primary focus of this work, we leave this to future work.

| | Time Elapsed (seconds) | Percentage | Peak VRAM Usage (MB) | Percentage |
|---|---|---|---|---|
| AnyEdit | 509.41 | 100.0% | 51249 | 100% |
| $\mu$KE w/ cache | 530.78 | 104.2% | 51719 | 100.9% |
| $\mu$KE w/o cache | 681.79 | 133.8% | 51719 | 100.9% |

Table 11: Computational overhead of $\mu$KE compared to AnyEdit.

### D.7 Case Study

#### D.7.1 *A failing example of AnyEdit's performance on EditEverything-Poetry*

> **Question**
>
> ```
> Write a poem inspired by the sound of rain.
> ```

> **Edit Target**
>
> ```
> The rain begins, a whispered thread,
> Upon the earth, so cool, widespread,
> A lullaby on rooftop's face,
> That soothes the heart in soft embrace.
>
> Each drop, a note, a pulse, a sigh,
> A rhythm drawn from weeping sky,
> Yet in its fall, a grace anew,
> As soil drinks deep the silver hue.
>
> O rain, you sing of earth's release,
> Of storms that break but yield to peace,
> And when the final echoes wane,
> The world begins to bloom again.
> ```

Below are two outputs from the edited Qwen2.5-7B-Instruct models using $\mu$KE and AnyEdit. Special tokens are omitted.

> **$\mu$KE**
>
> ```
> The rain begins, a whispered thread,
> Upon the earth, so cool, widespread,
> A lullaby on rooftop's face,
> That soothes the heart in soft embrace.
>
> Each drop, a note, a pulse, a sigh,
> A rhythm drawn from weeping sky,
> Yet in its fall, a grace anew,
> As soil drinks deep the silver hue.
>
> O rain, you sing of earth's release,
> Of storms that break but yield to peace,
> And when the final echoes wane,
> The world begins to bloom again.
> ```

> **AnyEdit**
>
> ```
> The rain begins, a whispered thread,
> Upon the earth, so cool, so spread,
> A lullaby on roof and leaf,
> That soothes the soul from grief to peace.
>
> Each drop, a note, in time arranged,
> A rhythm soft, both day and night
> displayed,
> It dances light on windowpane,
> A melody that eases pain.
>
> In heart it stirs, a gentle stir,
> Of memories, both old and new, no more,
> For every sound, a story told,
> Of seasons past, of days unrolled.
>
> So let me rest beneath this sound,
> And let my thoughts in peace be found,
> For in the rain's soft, soothing song,
> I find the calm I long for long.
> ```

