# OpenReview forum: "$\mu$KE: Matryoshka Unstructured Knowledge Editing of Large Language Models"
_colmweb.org/COLM/2025/Conference — COLM 2025_

### Official Review · Reviewer_LBJL · 2025-05-06

**Rating:** 6
**Confidence:** 4
**Ethics Flag:** 1

**Summary:**

1. This paper presents µKE, a new method for editing knowledge in LLMs. It fixes the problem of broken causal dependencies in existing window-based autoregressive methods by using a nested(Matryoshka-style) objective and adaptive optimization.
2. Tests show it outperforms current methods by up to 12.33% on UnKEBench-Ori-BLEU.

**Reasons To Accept:**

*  Provide a theoretical analysis of the issue regarding missing working memory dependency in AnyEdit.
* The proposed Matryoshka memory update mechanism is an elegant solution that effectively addresses these limitations while remaining compatible with the locate-and-edit paradigm.
* It's a reasonable improvement on anyEdit that delivers significant performance gains.

**Reasons To Reject:**

* The paper cites the wrong **AnyEdit** paper - it references an image editing paper (https://arxiv.org/pdf/2411.15738) instead of the correct knowledge editing one (https://arxiv.org/pdf/2502.05628). I think this is a very serious and low-level mistake, because anyEdit is a main comparison method in this paper.

* There's inconsistency in how experimental results are reported: the paper mixes results quoted directly from the AnyEdit paper (for MEMIT and AlphaEdit) with their own reproductions (for AnyEdit). The big differences between the reported AnyEdit results and the authors' reproduction need explanation. For example, on UnKEBench-Para-Rouge-L, AnyEdit reports 95.60±0.35 while this paper's reproduction shows only 80.51.

* The statistical significance is questionable: the paper only uses 3-run averages without reporting standard deviations, making it hard to judge if the improvements over AnyEdit (which includes error bars) are statistically reliable.

* Details about the datasets are missing: the paper doesn't say how many samples from each test set were used, which matters because editing quantity directly impacts performance. There's also no analysis of how performance changes with the number of edits.

* Should be some performance analysis comparing with AnyEdit, as calculating Matryoshka Memory requires relying on larger windows (more tokens), which will result in more time overhead.

* The hyperparameter analysis is limited: the paper doesn't analyze important parameters like window size and overlap settings, which significantly impact editing performance.

* There are formatting and writing issues: for example, "Matryoshka" is misspelled as "Matryoshak" in Figures 5 and 6.

---

> ### Author Response · Authors · 2025-06-03
>
> We thank reviewer for detailed and constructive questions.
>
> # Typos in the paper
> We apologize for this oversight. Crucially, all our experiments and comparisons used the knowledge-editing version of AnyEdit; this was a citation typo and does not affect any results or conclusions. We will fix this typo and conduct a thorough proofread of the manuscript to resolve any remaining formatting or writing issues.
> # Inconsistency between the reproduced numbers and the reported numbers in AnyEdit
> Our reproduced numbers are produced by the latest code from https://github.com/jianghoucheng/AnyEdit with default parameters. We hypothesize the difference might come from environment setups, as we found the provided environment configuration is incomplete in the repo and caused non-trivial errors. We had to upgrade PyTorch and Huggingface's transformers library according to a GitHub Issue. We will include a full list of packages required for reproduction.
>
> # Statistical reliability
> We run three trials for each experiment since we did not observe significant variance, as we report the standard deviations below.
>
> UnKEBench:
>
> |  | Ori. BLEU | Ori. BERT | Ori. R-L | Para. BLEU | Para. BERT | Para. R-L |
> | --- | --- | --- | --- | --- | --- | --- |
> | Llama, μΚΕ  | 90.24$\pm$0.16 | 97.70$\pm$0.04 | 91.75$\pm$0.07 | 76.72$\pm$0.08 | 93.78$\pm$0.05 | 77.59$\pm$0.13 |
> | Llama, μΚΕ* | 99.81$\pm$0.05 | 99.97$\pm$0.02 | 99.79$\pm$0.05 | 77.82$\pm$0.07 | 93.75$\pm$0.03 | 80.08$\pm$0.00 |
> | Qwen, μΚΕ  | 96.34$\pm$0.24 | 99.02$\pm$0.06 | 97.02$\pm$0.11 | 75.44$\pm$0.29 | 92.25$\pm$0.06 | 75.34$\pm$0.36 |
> | Qwen, μΚΕ* | 99.27$\pm$0.05 | 99.61$\pm$0.01 | 99.24$\pm$0.05 | 50.69$\pm$0.54 | 81.57$\pm$0.60 | 48.33$\pm$0.01 |
>
> CounterFact:
>
> |  | Ori. BLEU | Ori. BERT | Ori. R-L | Para. BLEU | Para. BERT | Para. R-L |
> | --- | --- | --- | --- | --- | --- | --- |
> | Llama, μΚΕ  | 94.72$\pm$0.16 | 98.75$\pm$0.03 | 93.76$\pm$0.89 | 41.22$\pm$0.26 | 63.72$\pm$0.14 | 37.40$\pm$0.12 |
> | Llama, μΚΕ* | 99.96$\pm$0.02 | 99.99$\pm$0.00 | 99.96$\pm$0.03 | 43.60$\pm$0.42 | 64.53$\pm$0.40 | 40.93$\pm$0.16 |
> | Qwen, μΚΕ  | 95.23$\pm$0.03 | 98.76$\pm$0.09 | 95.57$\pm$0.18 | 45.20$\pm$0.16 | 64.89$\pm$0.16 | 39.32$\pm$0.09 |
> | Qwen, μΚΕ* | 99.68$\pm$0.04 | 99.89$\pm$0.02 | 99.67$\pm$0.06 | 35.20$\pm$0.90 | 57.46$\pm$0.52 | 29.41$\pm$0.48 |
>
> MQuAKE:
>
> |  | Ori. BLEU | Ori. BERT | Ori. R-L |
> | --- | --- | --- | --- |
> | Llama, μΚΕ  | 94.72$\pm$0.16 | 98.75$\pm$0.03 | 93.76$\pm$0.89 |
> | Llama, μΚΕ* | 99.96$\pm$0.02 | 99.99$\pm$0.00 | 99.96$\pm$0.03 |
> | Qwen, μΚΕ  | 95.23$\pm$0.03 | 98.76$\pm$0.09 | 95.57$\pm$0.18 |
> | Qwen, μΚΕ* | 99.68$\pm$0.04 | 99.89$\pm$0.02 | 99.67$\pm$0.06 |
>
> # Datasets and number of edits.
>
> We followed AnyEdit and ran the full datasets for each benchmark: UnKEBench (1,000 samples), AKEW-CounterFact (975 samples), AKEW-MQuAKE (354 samples). Below are the batch editing results with different numbers of edits.
>
> UnKEBench:
>
> |  | Ori. BLEU | Ori. BERT | Ori. R-L | Para. BLEU | Para. BERT | Para. R-L |
> | --- | --- | --- | --- | --- | --- | --- |
> | AnyEdit*, 2 | 95.89 | 98.81 | 97.08 | 46.10 | 79.99 | 48.88 |
> | AnyEdit*, 4 | 95.90 | 98.78 | 97.06 | 44.18 | 79.12 | 47.11 |
> | AnyEdit*, 8 | 95.55 | 98.79 | 96.86 | 43.45 | 78.59 | 46.25 |
> | AnyEdit*, 16 | 93.34 | 98.18 | 95.28 | 41.79 | 77.52 | 44.50 |
> | AnyEdit*, 32 | 60.39 | 86.82 | 66.12 | 36.99 | 74.83 | 39.27 |
> | - | - | - | - | - | - | - |
> | μΚΕ*, 2 | 99.23 | 99.62 | 99.14 | 49.48 | 80.40 | 46.01 |
> | μΚΕ*, 4 | 99.17 | 99.58 | 99.18 | 49.29 | 80.14 | 45.67 |
> | μΚΕ*, 8 | 99.07 | 99.51 | 99.03 | 47.41 | 79.16 | 43.47 |
> | μΚΕ*, 16 | 98.00 | 99.32 | 98.06 | 46.28 | 78.51 | 42.48 |
> | μΚΕ*, 32 | 74.16 | 90.97 | 74.09 | 42.97 | 76.57 | 38.14 |
>
> CounterFact:
>
> |  | Ori. BLEU | Ori. BERT | Ori. R-L | Para. BLEU | Para. BERT | Para. R-L |
> | --- | --- | --- | --- | --- | --- | --- |
> | AnyEdit*, 2 | 96.67 | 99.08 | 97.42 | 30.60 | 55.85 | 31.46 |
> | AnyEdit*, 4 | 96.39 | 99.06 | 97.26 | 29.16 | 54.71 | 32.37 |
> | AnyEdit*, 8 | 96.51 | 99.09 | 97.34 | 27.98 | 54.43 | 33.06 |
> | AnyEdit*, 16 | 96.07 | 99.02 | 97.30 | 26.80 | 53.26 | 33.29 |
> | AnyEdit*, 32 | 93.02 | 98.29 | 94.86 | 26.89 | 52.54 | 32.22 |
> | - | - | - | - | - | - | - |
> | μΚΕ*, 2 | 99.69 | 99.85 | 99.59 | 33.89 | 55.43 | 28.65 |
> | μΚΕ*, 4 | 99.65 | 99.83 | 99.57 | 32.44 | 54.81 | 29.38 |
> | μΚΕ*, 8 | 99.64 | 99.84 | 99.60 | 31.73 | 54.52 | 30.29 |
> | μΚΕ*, 16 | 99.41 | 99.78 | 99.34 | 31.52 | 54.43 | 31.20 |
> | μΚΕ*, 32 | 97.84 | 99.38 | 97.92 | 31.34 | 54.74 | 30.50 |
>
> MQuAKE:
>
> |  | Ori. BLEU | Ori. BERT | Ori. R-L |
> | --- | --- | --- | --- |
> | AnyEdit*, 2 | 98.22 | 99.63 | 98.36 |
> | AnyEdit*, 4 | 97.90 | 99.51 | 98.24 |
> | AnyEdit*, 8 | 97.41 | 99.36 | 97.96 |
> | AnyEdit*, 16 | 96.66 | 99.21 | 97.59 |
> | AnyEdit*, 32 | 65.57 | 87.04 | 71.72 |
> | - | - | - | - |
> | μΚΕ*, 2 | 99.42 | 99.74 | 99.39 |
> | μΚΕ*, 4 | 99.30 | 99.73 | 99.28 |
> | μΚΕ*, 8 | 99.45 | 99.73 | 99.42 |
> | μΚΕ*, 16 | 98.14 | 99.28 | 98.19 |
> | μΚΕ*, 32 | 78.37 | 91.20 | 79.10 |

---

> > ### Author Response · Authors · 2025-06-03
> >
> > # Overhead
> > Please refer to our second point in the comment to Reviewer Nt4i.
> >
> > # Window size and overlap
> > During the rebuttal, we conduct additional experiments on the window sizes. We sample 100 editing data points and apply μΚΕ on Llama with different window sizes. We do not observe significant performance outliers.
> >
> > | Window Size | UnKEBench, Ori. BLEU | UnKEBench, Ori. BERT | UnKEBench, Ori. R-L | UnKEBench, Para. BLEU | UnKEBench, Para. BERT | UnKEBench, Para. R-L | MQuAKE, Ori. BLEU | MQuAKE, Ori. BLEU | MQuAKE, Ori. BLEU |
> > | --- | --- | --- | --- | --- | --- | --- | --- | --- | --- |
> > | 10 | 92.28 | 98.02 | 92.48 | 80.68 | 94.45 | 82.64 | 91.83, | 91.83 | 91.83 |
> > | 20 | 89.53 | 97.56 | 91.04 | 75.93 | 93.71 | 77.39 | 91.90 | 91.90 | 91.90 |
> > | 30 | 93.09 | 98.06 | 93.21 | 79.53 | 93.69 | 80.45 | 91.74 | 91.74 | 91.74 |
> > | 40 | 89.85 | 97.81 | 92.23 | 78.44 | 93.37 | 78.76 | 92.79 | 92.79 | 92.79 |
> > | 50 | 91.34 | 97.56 | 91.78 | 77.72 | 93.46 | 80.27 | 92.40 | 92.40 | 92.40 |
> > | 60 | 93.25 | 98.42 | 94.59 | 79.61 | 94.05 | 80.24 | 94.62 | 94.62 | 94.62 |
> > | 70 | 91.62 | 97.81 | 92.97 | 81.14 | 94.06 | 81.31 | 91.67 | 91.67 | 91.67 |
> > | 80 | 96.39 | 99.11 | 96.70 | 82.16 | 94.97 | 82.76 | 91.55 | 91.55 | 91.55 |
> > | 90 | 93.07 | 98.57 | 92.75 | 76.75 | 92.79 | 78.38 | 92.61 | 92.61 | 92.61 |
> > | 100 | 89.62 | 96.70 | 89.58 | 73.78 | 91.25 | 73.33 | 92.61 | 92.61 | 92.61 |
> >
> > We would like to clarify that overlap is by design included in the matryoshka objective, as illustrated in Figure 2 of our manuscript.

---

> > > ### Comment · Reviewer_LBJL · 2025-06-06
> > >
> > > I appreciate the authors' effort in conducting additional evaluations within a short timeframe. However, my primary concerns remain: (1) the necessity of using the BLEU metric (as the discrepancies between the reproduced results in this paper and the original AnyEdit results are not significant if BLEU is excluded), and (2) the lack of a thorough explanation for the inconsistencies in the results (with pre-edited, MEMIT, and AlphaEdit referring to those reported in prior work, while AnyEdit is based on the authors' own reproduction). Therefore, I believe my current score is appropriate.

---

> > > > ### Author Response · Authors · 2025-06-08
> > > >
> > > > # BLEU
> > > >
> > > > We include BLEU for these reasons.
> > > >
> > > > (a) BLEU is a very commonly used metric in the NLP tasks, including those that alter a model’s output, such as [1, 2, 3].
> > > >
> > > > (b) BLEU, in addition to BERTScore and ROUGE, was included in the UnKE paper and also reported in the codebase of AnyEdit.
> > > >
> > > > (c) Our experiments have shown that the matryoshka style update significantly improves BLEU, which penalizes extra or missing tokens. We believe it is important when factual edits hinge on precise phrases.
> > > >
> > > > [1] Wei, et al. Sentence-Level or Token-Level? A Comprehensive Study on Knowledge Distillation, *IJCAI 2024*
> > > >
> > > > [2] Subramani, et al. Extracting Latent Steering Vectors from Pretrained Language Models, *ACL 2022 Findings*
> > > >
> > > > [3] Guo, et al. Steering Large Language Models for Cross-lingual Information
> > > > Retrieval, *SIGIR 2024*
> > > >
> > > > # Baseline reproduction
> > > >
> > > > Using the same codebase from AnyEdit (https://github.com/jianghoucheng/AnyEdit), we reproduce the results of pre-edited, MEMIT, AlphaEdit, and UnKE and report as follows. We will make sure all of our results are reproduced to avoid inconsistencies in our manuscript.
> > > >
> > > > On UnKEBench:
> > > >
> > > > |  | Ori. BLEU | Ori. BERT | Ori. R-L | - | Para. BLEU | Para. BERT | Para. R-L |
> > > > | --- | --- | --- | --- | --- | --- | --- | --- |
> > > > | Llama, pre-edited | 20.09 | 71.38 | 25.49 | - | 19.80 | 71.29 | 25.31 |
> > > > | Llama, MEMIT | 24.97 | 76.21 | 30.03 | - | 22.51 | 74.50 | 28.29 |
> > > > | Llama, AlphaEdit | 21.43 | 73.76 | 26.65 | - | 20.38 | 72.85 | 25.88 |
> > > > | Llama, UnKE | 93.19 | 98.07 | 92.92 | - | 79.19 | 93.37 | 79.16 |
> > > > | - | - | - | - | - | - | - | - |
> > > > | Qwen, pre-edited | 20.92 | 72.58 | 27.64 | - | 20.51 | 72.18 | 26.60 |
> > > > | Qwen, MEMIT | 45.18 | 78.10 | 38.21 | - | 40.89 | 76.63 | 34.19 |
> > > > | Qwen, AlphaEdit | 49.75 | 80.57 | 42.93 | - | 45.37 | 78.41 | 38.42 |
> > > > | Qwen, UnKE | 91.70 | 96.82 | 90.77 | - | 57.55 | 84.10 | 51.97 |
> > > >
> > > > On AKEW-CounterFact:
> > > >
> > > > |  | Ori. BLEU | Ori. BERT | Ori. R-L | - | Para. BLEU | Para. BERT | Para. R-L |
> > > > | --- | --- | --- | --- | --- | --- | --- | --- |
> > > > | Llama, pre-edited | 13.89 | 68.33 | 18.76 | - | 15.81 | 42.22 | 13.18 |
> > > > | Llama, MEMIT | 31.88 | 75.83 | 31.23 | - | 17.52 | 47.27 | 15.88 |
> > > > | Llama AlphaEdit | 23.52 | 72.54 | 24.98 | - | 16.39 | 45.13 | 14.07 |
> > > > | Llama, UnKE | 98.20 | 99.56 | 98.15 | - | 40.70 | 61.09 | 36.04 |
> > > > | - | - | - | - | - | - | - | - |
> > > > | Qwen, pre-edited | 28.15 | 69.66 | 20.98 | - | 29.54 | 52.15 | 19.31 |
> > > > | Qwen, MEMIT | 45.08 | 77.16 | 38.95 | - | 32.99 | 56.15 | 25.80 |
> > > > | Qwen, AlphaEdit | 49.88 | 80.42 | 45.35 | - | 34.65 | 56.94 | 27.74 |
> > > > | Qwen, UnKE | 92.07 | 97.68 | 91.42 | - | 38.38 | 58.92 | 29.16 |
> > > >
> > > >
> > > > On AKEW-MQuAKE:
> > > >
> > > > |  | Ori. BLEU | Ori. BERT | Ori. R-L |
> > > > | --- | --- | --- | --- |
> > > > | Llama, pre-edited | 18.84 | 69.79 | 21.16 |
> > > > | Llama, MEMIT | 26.61 | 69.93 | 25.75 |
> > > > | Llama AlphaEdit | 22.57 | 69.78 | 22.74 |
> > > > | Llama, UnKE | 95.11 | 98.94 | 94.58 |
> > > > | - | - | - | - |
> > > > | Qwen, pre-edited | 24.12 | 69.05 | 21.03 |
> > > > | Qwen, MEMIT | 41.54 | 71.61 | 35.46 |
> > > > | Qwen, AlphaEdit | 45.14 | 72.76 | 39.83 |
> > > > | Qwen, UnKE | 88.11 | 95.19 | 86.45 |

---

### Official Review · Reviewer_Nt4i · 2025-05-10

**Rating:** 7
**Confidence:** 4
**Ethics Flag:** 1

**Summary:**

This paper addresses the task of unstructured knowledge editing for LLMs and introduces a novel method called Matryoshka Unstructured Knowledge Editing ($\mu$KE). The authors identify a key limitation in current state-of-the-art approaches, such as AnyEdit, which relies on a window-based autoregressive editing paradigm that fails to preserve memory dependency—i.e., the influence of early edits on subsequent token generation is lost. To remedy this, $\mu$KE introduces a Matryoshka-style objective that enforces progressive memory-to-output alignment by nesting window targets, and incorporates adaptive loss coefficients based on the gradient affinity between target segments, thereby preserving causal dependencies throughout sequential memory updates.

**Reasons To Accept:**

- The paper provides a theoretical analysis of the limitations of the window-based editing strategy by contrasting its gradient flow with that of parallel optimization. In particular, Equation (9) reveals the missing cross-window gradients and formally supports the need for encoding memory dependency, offering a theoretical foundation for the proposed method.
- $\mu$KE demonstrates greater optimization stability and robustness compared to existing baselines. Through a systematic analysis (Figures 5 and 6), the authors show that $\mu$KE maintains higher edit efficacy across varying target lengths and optimization steps, which is especially beneficial for long-form unstructured editing tasks.

**Reasons To Reject:**

- The proposed method exhibits a drop in locality performance on certain metrics (e.g., IFEval on Qwen2.5), indicating that the model’s ability to preserve general capabilities remains sensitive to editing. Could you attempt to explain the possible reasons for this phenomenon?
- Compared to AnyEdit, $\mu$KE introduces more intricate components, such as multi-target objectives and iterative gradient affinity calculations for $λ_{i,j}$, yet the paper lacks an explicit evaluation of computational overhead in terms of runtime, memory usage, or scalability.
- Baseline comparison is somewhat limited.The paper compares $\mu$KE primarily against MEMIT, UnKE, and AnyEdit. However, it does not include recent strong baselines or lifelong editing frameworks, which weakens the generality of its conclusions.
- While the derivations in Section 3.1 are sound, the inclusion of long gradient expansions in the main body hinders readability. These technical details could be consolidated or moved to the appendix to improve narrative flow and focus.

---

> ### Author Response · Authors · 2025-06-03
>
> We appreciate the reviewer’s thoughtful comments and address them below.
>
> # Potential reason for locality performance drop on certain metrics
>
> We think this issue might be related to the granularity of localization. Note that AnyEdit and μΚΕ inherit pre-defined editing layer selection from MEMIT and UnKE, where we consider all working memory shifts happen on the same layer(s). Intuitively, an unstructured editing target, which can be long, can encompass knowledge of different levels, which likely requires more fine-grained designs in localization and corresponding updates. Due to the complexity of the designs, we leave this to future work.
>
> # Computational overhead
>
> We see a ~4.2% increase in time elapsed running with cache and ~33.8% without cache, which includes IO costs and intermediate inference, among others. We observe a negligible peak VRAM usage increase. Since efficiency is not the primary focus of this work, we leave this to future work.
>
> |  | Time Elapsed (seconds) | Percentage | Peak VRAM Usage (MB) | Percentage |
> | --- | --- | --- | --- | --- |
> | AnyEdit | 509.41  | 100.0% | 51249  | 100% |
> | μΚΕ (w/o mat cache) | 530.78  | 104.2% | 51719  | 100.9% |
> | μΚΕ (w/ mat cache) | 681.79  | 133.8% | 51719  | 100.9% |
>
> # Limited baseline comparison, lacking lifelong editing frameworks.
>
> We perform additional experiments with a recent lifelong editing baseline, WISE [1], and adopt code and hyperparameters from EasyEdit [2]. We use Llama and Qwen as shortcuts to `Llama3-8B-Instruct` and `Qwen2.5-7B-Instruct`. We observe incomparable and unstable performance of WISE compared to AnyEdit and μΚΕ under the default settings.
>
> WISE on UnKEBench:
>
> |  | Ori. BLEU | Ori. BERT | Ori. R-L | Para. BLEU | Para. BERT | Para. R-L |
> | --- | --- | --- | --- | --- | --- | --- |
> | Llama | 17.14 | 77.90 | 62.75 | 16.18 | 74.64 | 55.35 |
> | Qwen | 24.64 | 78.15 | 55.42 | 25.70 | 77.29 | 52.81 |
>
> On CounterFact:
>
> |  | Ori. BLEU | Ori. BERT | Ori. R-L | Para. BLEU | Para. BERT | Para. R-L |
> | --- | --- | --- | --- | --- | --- | --- |
> | Llama | 18.04 | 86.79 | 93.76 | 13.73 | 66.21 | 53.95 |
> | Qwen | 32.80 | 73.23 | 31.37 | 30.96 | 54.38 | 25.88 |
>
> On MQuAKE:
>
> |  | Ori. BLEU | Ori. BERT | Ori. R-L |
> | --- | --- | --- | --- |
> | Llama | 20.76 | 87.26 | 94.03 |
> | Qwen | 35.46 | 77.30 | 46.16 |
>
> On EditEverything:
>
> |  | Math, BERT | Math, R-L | Code, BERT | Code, R-L | Poetry, BERT | Poetry, R-L | News, BERT | News, R-L | Chemistry, BERT | Chemistry, R-L |
> | --- | --- | --- | --- | --- | --- | --- | --- | --- | --- | --- |
> | Llama | 90.39 | 61.58 | 84.92 | 87.12 | 90.19 | 91.96 | 87.83 | 75.89 | 97.25 | 92.15 |
> | Qwen | 94.38 | 72.73 | 77.34 | 70.61 | 70.51 | 41.87 | 74.55 | 40.49 | 92.76 | 65.51 |
>
>
> # Technical details could be consolidated or moved to the appendix to improve narrative flow and focus.
>
> Thank you for your suggestion; we will simplify this part and move the details to the Appendix.
>
>
>
> [1] Wang, Peng , et al. “WISE: Rethinking the Knowledge Memory for Lifelong Model Editing of Large Language Models.” *NeurIPS 2024*
>
> [2] https://github.com/zjunlp/EasyEdit

---

### Official Review · Reviewer_tkzF · 2025-05-12

**Rating:** 7
**Confidence:** 4
**Ethics Flag:** 1

**Summary:**

This paper addresses a long-form knowledge editing, by improving AnyEdit, which takes a window-by-window method by splitting down the original long edit target into windows, while this method overlooks the memory dependency. Then, this paper proposes a matryoshka-style working memory, by distributing target figures in a balanced manner, theoretically deriving from a full loss function. The experiment results show the proposed Matryoshka-style editing leads to improvements over AnyEdit across two LLM settings across long-term knowledge editing tasks.

**Reasons To Accept:**

-	The motivation is largely clear and the proposed method is novel.
-	The experiment results are positive, leading to improvements over AnyEdit

**Reasons To Reject:**

-	More ablation study to validate the selection of coefficients of Eq. (12) is required.

---

> ### Author Response · Authors · 2025-06-03
>
> Thank you for the supportive review!
>
> # More Ablation Studies on the selection of coefficients in Eq.(12)
>
> We conduct ablation studies on the selection of coefficients during the rebuttal period. Due to the time constraint, we ablate on the vanilla matryoshka (with all coefficients set to 1), and coefficients computed by figure affinity with different steps of optimization. We show μΚΕ results based on Qwen, 100 samples in the following table. Results show that vanilla matryoshka and that of adaptive coefficients are effective in edit success, yet adaptive coefficients help in improving generality of the edit (para. results), which is consistent with our previous finding in Figure 6 in the original paper.
>
> ---
>
> |  | CF, Ori. BLEU | CF, Ori. BERT | CF, Ori. ROUGE-L | CF, Para. BLEU | CF, Para. BERT | CF, Para. ROUGE-L |
> | --- | --- | --- | --- | --- | --- | --- |
> | matryoshka (baseline) | 96.25 | 98.70 | 96.83 | 41.91 | 63.10 | 37.56 |
> | adaptive, step=1 | 96.98 | 99.49 | 97.00 | 44.83 | 65.73 | 40.95 |
> | adaptive, step=2 | 96.68 | 99.56 | 97.04 | 45.16 | 65.93 | 41.00 |
> | adaptive, step=3 | 96.03 | 99.19 | 96.21 | **45.66** | **66.82** | **41.50** |

---

### Official Review · Reviewer_bMyA · 2025-05-13

**Rating:** 6
**Confidence:** 4
**Ethics Flag:** 1

**Summary:**

The paper introduces Matryoshka Unstructured Knowledge Editing (μΚΕ), a novel approach for editing the internal knowledge of Large Language Models (LLMs). This method aims to improve upon existing techniques, particularly window-based autoregressive methods like AnyEdit, by addressing the disruption of causal dependency between early memory updates and later output tokens. μΚΕ employs a Matryoshka-style objective and adaptive loss coefficients to preserve these dependencies. The authors provide a theoretical analysis of the limitations in current methods and demonstrate empirically that μΚΕ shows improved edit efficacy by up to 12.33% over state-of-the-art methods across two models and four benchmarks. The paper also highlights μΚΕ's robustness when applied to diverse formatted edits.

**Questions To Authors:**

Some suggestions:
- It's better to give a proper definition of working memory with some reference.

**Reasons To Accept:**

- The paper proposes a clear improvement on existing methods like AnyEdit by introducing the Matryoshka loss computation. This novel mechanism demonstrably achieves better performance in unstructured knowledge editing, as evidenced by the empirical evaluations
-  The authors provide a solid theoretical analysis of the limitations of current window-based autoregressive editing strategies, specifically the overlooking of memory dependency. The motivation for μΚΕ is well-grounded in addressing these identified shortcomings, particularly in preserving causality between edited memory and subsequent token distributions.

**Reasons To Reject:**

The primary concern with this work lies in the dataset and evaluation methodology, specifically concerning the nature of "unstructured editing" and the definition of a successful edit.
I agree that unstructured editing beyond original fact-level editing is important for the model editing area.
In the fact-level edit, the model should give us the correct fact. However, for the cases shown in the paper, I think directly asking the model to generate the exact output as the given golden answer is not necessary.
For example, the case the author showed in the appendix is
> 'Write a poem inspired by the sound of rain.'

I think the AnyEdit output is also acceptable. For these inputs, I don't think we need the model to give us the exact token match.
The emphasis on exact output replication could inadvertently encourage models to memorize training data. This raises potential issues related to overfitting and privacy leakage, which are generally undesirable characteristics in LLMs. The goal of editing should arguably be to instill or correct knowledge and behaviors in a more generalized way, rather than to force specific phrasings for specific inputs.
The unstructured editing should be things like evaluating the inconsistent content, conflicting information, or hallucinations.

Hence, the method has its contribution, but for the task, I think the author should have a deep discussion here.

---

> ### Author Response · Authors · 2025-06-02
>
> We sincerely thank the reviewer for the constructive review.
> ## Clarification on unstructured editing dataset and evaluation
>
> We would like to first clarify about the datasets and evaluation setups we adopted for unstructured editing.
>
> - We follow the evaluation setups in existing unstructured editing studies [1, 2]. We evaluated our approach on most existing benchmarks from previous work, including diverse types of knowledge editing (counterfactual, multi-hop) and diverse formats of knowledge (news, chemistry, etc.).
> - The poem case study is just an illustration of the capability of our improved method in achieving higher success and more accurate edit on the lexical level. We agree with the reviewer that certain benchmarks might not be the most appropriate for evaluating unstructured editing. We will follow the reviewer's advice to add a discussion in the paper for the proper selection of benchmarks.
>
> In the rebuttal period, we conduct another experiment on the hallucination reduction benchmark from SelfCheckGPT [3], which is a more realistic scenario. As shown in the following table, μΚΕ consistently outperforms AnyEdit and achieves a reasonably high score for hallucination reduction.
>
> |  | BLEU | BERTScore | Rouge-L |
> | --- | --- | --- | --- |
> |Pre-edit | 10.58 | 42.13 | 21.93 |
> | AnyEdit | 66.59 | 82.64 | 84.66 |
> | μΚΕ | 86.77 | 91.80 | 87.17 |
>
>
> ## The definition of “working memory”
>
> The “working memory” here indicate the specific hidden states corresponding to the located layers that are hypothesized to contain the knowledge to be edited and can be updated with a bias term using gradient descent. Conceptually, this can be connected to definition in human working memory in psychology [4] and cognitive neuroscience [5, 6]. In short, both in human cognition and in the μΚΕ approach, “working memory” acts as a short-lived, capacity-limited store that is actively manipulated (through attention or gradient steps) to change future outcomes (behavioral response or token probabilities). The cognitive-science emphasis on neural circuits in prefrontal/parietal areas echoes the μΚΕ reliance on a specific transformer layer ($l_\text{WM}$) whose activations effectively serve the same functional purpose: keeping relevant information “online” so that it can be selectively updated and then used to guide subsequent processing.
>
> [1] Deng, Jingcheng, et al. "Everything is Editable: Extend Knowledge Editing to Unstructured Data in Large Language Models." ICLR 2025
>
> [2] Jiang, Houcheng, et al. "AnyEdit: Edit Any Knowledge Encoded in Language Models." ICML 2025
>
> [3] Manakul, Potsawee, Adian Liusie, and Mark Gales. "SelfCheckGPT: Zero-Resource Black-Box Hallucination Detection for Generative Large Language Models." *EMNLP 2023*
>
> [4] Hitch, Graham James, and A. D. Baddeley. "Verbal reasoning and working memory." *The Quarterly Journal of Experimental Psychology* 28.4 (1976): 603-621.
>
> [5] Finn, Emily S., et al. "Layer-dependent activity in human prefrontal cortex during working memory." *Nature neuroscience* 22.10 (2019): 1687-1695.
>
> [6] Degutis, Jonas Karolis, et al. "Dynamic layer-specific processing in the prefrontal cortex during working memory." *Communications biology* 7.1 (2024): 1140.

---

> > ### Comment · Reviewer_bMyA · 2025-06-03
> >
> > Thanks for your reply. I will raise the score, and I hope the author discusses more about the evaluation in the paper.

---

### Decision · Program_Chairs · 2025-07-08

**Decision:**

Accept

**Comment:**

**Pros**

Quality:
- The quality of the paper and its experiments are well-motivated and clearly defined. The additional ablation study is also informative.
- The experiments demonstrate the proposed model's strong performance compared to the baseline, especially with varying lengths.

Clarity:
- The paper is generally well-written and easy to understand.

Originality:
- The proposed method is both novel and effective.

Significance:
- The experimental results show a positive improvement over baseline methods. This method could have a significant impact on unstructured text editing.

**Cons**

Quality:
- The paper has a typo, citing the wrong reference for AnyEdit, the main method that they compared with.
- As Reviewer LBJL points out, the experiment numbers should be more precise and verified. The authors should include the results from the author's response to the updated version of the paper.
- The presented results are somewhat inconsistent with those in the original paper. The authors should double-check the numbers and provide an explanation for any discrepancies.

Clarity:
- Some sections of the paper should be moved to the appendix to improve readability.

Significance:
- The experimental setting for unstructured knowledge editing is ill-defined. The authors should acknowledge the limitations of this experimental setup.

Overall, reviewers agree that the proposed method is novel, sound, and demonstrates a substantial performance improvement over baseline methods. However, there are typos and writing suggestions from the review that should be addressed. Additionally, more experiment details, as well as an explanation for the experiment number discrepancies noted by Reviewer LBJL, should be included. The suggested modifications are all rather minor and could be achieved in the updated version of the paper.